# Benchmark of Rotor Position Sensor Technologies for Application in Automotive Electric Drive Trains

**Christoph Datlinger *** and **Mario Hirz**

Institute of Automotive Engineering, Graz University of Technology, 8010 Graz, Austria; mario.hirz@tugraz.at

\* Correspondence: christoph.datlinger@tugraz.at; Tel.: +43-316-873-35264

**Abstract:** Rotor shaft position sensors are required to ensure the efficient and reliable control of Permanent Magnet Synchronous Machines (PMSM), which are often applied as traction motors in electrified automotive powertrains. In general, various sensor principles are available, e.g., resolvers and inductive- or magnetoresistive sensors. Each technology is characterized by strengths and weaknesses in terms of measurement accuracy, space demands, disturbing factors and costs, etc. Since the most frequently applied technology, the resolver, shows some weaknesses and is relatively costly, alternative technologies have been introduced during the past years. This paper investigates state-of-the-art position sensor technologies and compares their potentials for use in PMSM in automotive powertrain systems. The corresponding evaluation criteria are defined according to the typical requirements of automotive electric powertrains, and include the provided sensor accuracy under the influence of mechanical tolerances and deviations, integration size, and different electrical- and signal processing-related parameters. The study presents a mapping of the potentials of different rotor position sensor technologies with the target to support the selection of suitable sensor technologies for specified powertrain control applications, addressing both system design and components development.

**Keywords:** automotive electric powertrain; permanent magnet synchronous motor; rotor position sensor; resolver; inductive position sensor; eddy current position sensor; Hall sensor; magnetoresistive position sensor

## 1. Introduction

The electrification of vehicles is becoming increasingly widespread in order to reduce greenhouse gas emissions and fulfill the corresponding exhaust emission legislations. Regardless of the electric powertrain architecture, e.g., Hybrid Electric Vehicle (HEV) or Battery Electric Vehicle (BEV), there are two main types of traction motors used today: Induction Motors (IM) and Permanent Magnet Synchronous Motors (PMSM). The control strategies differ significantly according to the selected motor type [1]. Thus, feedback for controlling the electric machine, which is delivered by a rotor shaft sensor, can be separated into two signal types: the rotor speed signal when applying an IM and the rotor position information when utilizing a PMSM. Sensorless control, as it is often found in the literature [2,3], is not used in automotive powertrains due to high demands on control reliability and is therefore not discussed further in detail here.

In general, the automotive industry sets high demands on the control of electrical drive trains, which are required for the correct electronically controlled commutation of these motor types. Accurate rotor speed and position information is vital to obtain precise torque and speed control. This enables the best motor efficiency, resulting in increased driving ranges, increased comfort (by reducing torque ripple) and maintaining Functional Safety (FUSA)-relevant issues, e.g., according to the ISO 26262 [4–6]. In addition, real-time information about rotor speed and position is required for a defined starting

direction from vehicle standstill and the prevention of the powertrain from unintended blocking. As an example, Automotive Safety Integrity Level-D (ASIL level-D) applications, which refer to the highest classification of injury risk and the most stringent level of safety measures, define a maximum position sensing angle error of 2° [7].

Contrary to the IM, the PMSM shows better efficiency and is therefore in widespread use in state-of-the-art automotive propulsion systems. Owing to this fact, the present paper focusses on the analysis of rotor position measurement technologies for PMSM-based automotive drivetrains. In general, different rotor shaft position sensor technologies are available for PMSM, whereby the selection of a suitable system is influenced by several factors, such as accuracy demands; sensibility to mechanical, magnetic and electrical disturbance; integration size; costs; etc. Investigations from the course of this research show that currently applied rotor position sensors in PMSM are resolver-types, inductive/eddy current-based sensors, magnetoresistive sensors or encoders. However, the latter type is not applied in automotive powertrains, since a high accuracy can only be achieved with high (expensive) effort.

In today's electric powertrain development processes, the sensor specifications delivered by sensor manufacturers are taken into account [8], but these do not provide sufficient details about the sensor characteristics. This often leads to a challenge for the automotive industry when selecting an unbiased sensor. In this context, the present work delivers detailed investigations of typical rotor position sensor technologies to provide a comparison of the respective sensor systems. A comprehensive evaluation of the sensor characteristics was performed regarding the angle accuracy under a variety of mechanical installation tolerances, operating temperatures and driving profiles (i.e., rotary speed variances). The aforementioned research activities are performed with the aid of a unique sensor test bench, which enables a highly accurate gauging of objects under test. Furthermore, important selection criteria for the drivetrain development, such as the sensor housing size and sensitivity to magnetic fields, costs, electrical supply effort and signal processing effort, were evaluated. Consequently, a holistic overview of state-of-the-art rotor position sensor system potentials is provided, including an analysis of the provoked error characteristics. The results of the study support the selection of sensor technology in terms of the described influencing factors for a given drivetrain application.

The paper is organized as follows. Section 2 provides a general overview of different state-of-the-art rotor position sensors for both automotive electric powertrains and auxiliaries, including a listing of general strengths and weaknesses of each sensor technology. In conjunction with the description of each sensor principle, a representative automotive application was selected to provide boundary conditions for the application of the different sensor technologies to be evaluated. In Section 3, a sensor test bench and measurement plan are introduced to deliver an overview of both the performed testing procedure and the data acquisition technologies. The measurement results of the different sensor technologies are presented in Section 4. The evaluation results are discussed in Section 5, which also includes a holistic comparison of the measurement results and an assessment of the potentials of the different sensor principles. Finally, conclusions are given in Section 6.

## 2. Automotive Rotor Position Sensors

Rotor position sensors are used in various fields of application in the automotive industry, e.g., in electric powertrain systems, electric cooling fans or pumps. As stated in [9], position measurement in automotive applications can be performed by different approaches. In general, contacting and non-contacting technologies come into use. An exemplary contacting-based angular measurement principle represents the electronic accelerator pedal (E-GAS), which is comprised of a potentiometer using wipers sliding on copper tracks. Such sensor architectures suffer from wear in long term use, and do not enable angle determination greater than 360°. Thus, this principle is not appropriate for the focused investigation object, which is an electric powertrain that requires non-contacting rotor position measurement. In what follows, state-of-the-art automotive non-contacting sensor principles are introduced and discussed, which are investigated in the scope of this work. These technologies

include resolver-, inductive-/eddy current-, Hall effect-based- and magnetoresistive-position sensors. Figure 1 represents a graphical overview of the four main types.

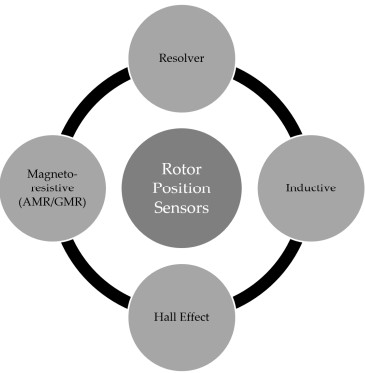

**Figure 1.** Considered automotive non-contacting rotor position sensor principles.

## 2.1. Principle of the Resolver Position Sensor

The state-of-the-art position sensing technology in automotive traction systems is the resolver, due to a great reliability in terms of rough conditions, for example contamination, temperature, shock and vibrations. The resolver appears in two types: Wound Field (WF) and Variable Reluctance (VR) resolvers. The difference lies in the design of the sensor's rotor. The WF resolver has a sinusoidal distributed excitation winding with a constant Air Gap (AG) [10]. In contrast, the VR resolver has a uniform distributed excitation winding, in combination with a sinusoidal-shaped AG rotor design [11]. Since the VR resolver is the most established resolver type in automotive drivetrain architectures, the present work focusses on this type. The discussed VR resolver consists of three coils: one excitation coil and two sensing coils. The excitation coil operates as a transmitter. Therefore, the resolver can be considered as a rotating electric transformer. The excitation signal itself is a high frequency sinusoidal voltage $U_{exc}$ at typically $f$ = 10 kHz. The generated electromagnetic flux is coupled through the shape of the rotor, which consists of a stamped ferromagnetic material. The sensing coils induce an electrical voltage comprising the excitation signal and a rotor position-dependent magnitude. Note that both sensing coils are mechanically displaced to deliver both a sine- and a cosine-shaped output signal while the rotor is in motion. Consequently, the sensor provides two Amplitude Modulated (AM) signals, which are shifted by 90°. To determine the angular information $\theta$, the excitation term must be removed. This process is performed in a specific Resolver-to-Digital Converter (RDC). Comprehensive study of this sensor system has been performed in previous works, see [12–14]. Figure 2 depicts the operating principle of a VR resolver including an exemplary sensor, which is utilized in automotive drivetrains of BEVs.

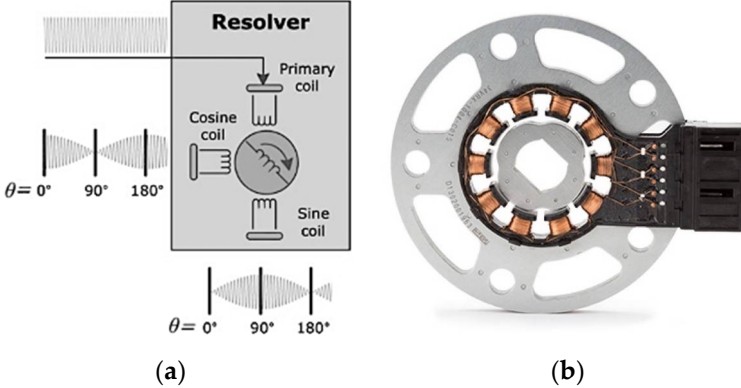

(**a**)                                          (**b**)

**Figure 2.** Variable Reluctance (VR) resolver position sensor. (**a**) Operating principle [13]; (**b**) Exemplary sensor for Battery Electric Vehicles (BEVs), reproduced from [15]. Copyright 2000–2020 MinebeaMitsumi Inc.

Hereafter, the principle is formulated mathematically by Equations (1)–(4). The utilized variables are described as follows. *A* represents the amplitude of the respective voltage. The resolver can be considered as a rotating transformer; thus, the feedback voltages $U_{sin}$ and $U_{cos}$ are reduced by a certain factor, which is denoted as transformation ratio *k*. The electrical rotor shaft position $\theta$ is determined by an arctangent function of the demodulated signals $V_{sin,demod}$ and $V_{cos,demod}$, which results in an extraction of the excitation term.

$$V_{exc} = A \times \sin(\omega t) \tag{1}$$

$$V_{sin} = kA \times \sin(\omega t) \times \sin(\theta) \tag{2}$$

$$V_{cos} = kA \times \sin(\omega t) \times \cos(\theta) \tag{3}$$

$$\theta = \arctan(\frac{V_{sin,demod}}{V_{cos,demod}}) \tag{4}$$

The resulting electrical angle is usually multiplied by a factor, which describes the number of poles. For electric powertrains, the number of poles of the resolver is related to the number of poles of the traction motor. A higher number of resolver poles increases the accuracy of the sensor, but also increases the costs, since a higher number of windings is required. These multi-speed resolvers are preferred for applications with high requirements on sensor system performance, e.g., electric drivetrains.

In the automotive sector, a resolver position sensor system is most commonly used for controlling PMSM-based powertrains. This is mainly because of the robustness and high accuracy of the technology. In the following, the state-of-the-art architecture of a resolver-based rotor position system and the corresponding signal processing are described. As shown in Figure 3, it consists of the resolver itself, a carrier generation unit, which comprises a digital stage and an analog amplification circuit, an Analog-to-Digital Converter (ADC) for digitizing the resolver feedback signals, a demodulation stage and an angle computation algorithm.

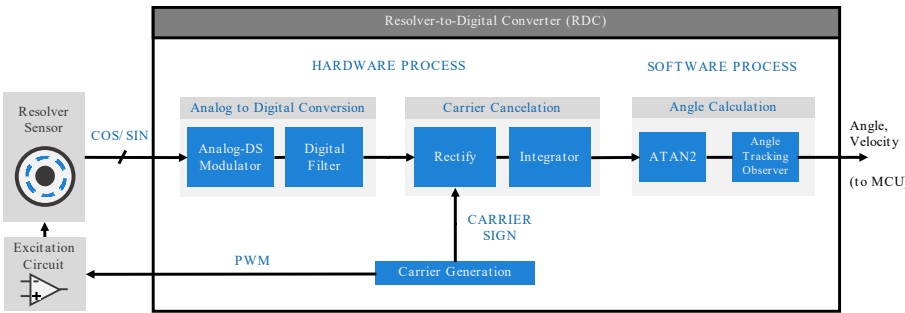

**Figure 3.** Exemplary illustration of a resolver rotor position signal processing system.

All signal processing stages are integrated in the RDC, which can be either a separate module or be integrated in the powertrain's ECU. The sensor is electrically connected differentially for both the excitation and the feedback signals using Twisted Pair (TP) cables to eliminated external interferences. The angular information is passed on by an interface, e.g., a Transistor Transistor Logic (TTL) encoder emulation, an Inter-Integrated Circuit ($I^2C$) bus when the RDC unit is placed closely to the subsequent Motor Control Unit (MCU) on a single PCB, or a CAN bus in case of a distributed system architecture. A detailed functional description of the individual processing stages can be found in [12,14]. Table 1 outlines the strengths and weaknesses of state-of-the-art VR resolver position sensors.

**Table 1.** Property summery of VR resolvers.

| | Attributes |
|---|---|
| Advantages | High accuracy and resolution |
| | Reliable |
| | Robust |
| Disadvantages | High costs |
| | High weight and inertia (speed limitation) |
| | Relatively large installation space |
| | High power consumption due to excitation |
| | Complex signal processing required for angle determination (i.e., RDC) |
| | Sensitiveness to mechanical tolerances e.g., eccentricity |
| | Not immune against stray fields |

### 2.2. Inductive Position Sensor

An alternative to the resolver position sensor is the inductive sensor principle, also denoted as an eddy current position sensor [8]. This sensor principle comprises two main components; a metallic rotor that can be of copper or aluminum and a coil design, which is printed on a two-sided Printed Circuit Board (PCB). Angular position determination, including signal processing, is computed on a dedicated sensor chip, which is also mounted on the same PCB [16]. Figure 4 depicts an exemplary inductive sensor application, including relevant components.

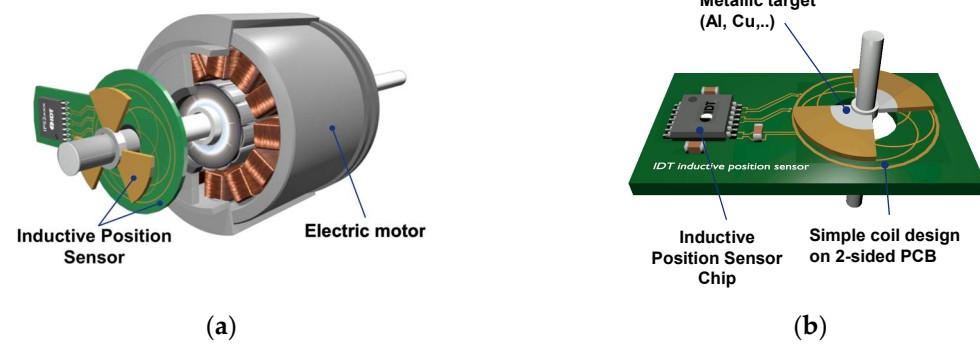

**Figure 4.** Inductive position sensor, reproduced from [16]. Copyright 2019, IDT. (**a**) Illustration of a through-shaft integration; (**b**) Relevant sensor components scheme.

As evident in the figure above, the inductive sensor can be integrated as a through-shaft design in the same manner as the formerly introduced resolver. It should be mentioned that, for the inductive sensor, other designs are feasible, e.g., end-of-shaft (EoS) or side-shaft (i.e., arc design). This makes this sensor technology flexible for a wide range of applications in electric vehicles and auxiliaries.

The working principle of inductive sensors is based on two fundamental physical principles: Michael Faraday's law of induction, and the effects of eddy currents, which have been discovered by Léon Foucault [16,17] (Michael Faraday (1791–1867): scientist in the field of electromagnetism and electrochemistry [18]; Léon Foucault (1819–1868): French physicist who discovered eddy currents and the effect of the earth's rotation [19]). The stator of the sensor (i.e., the PCB) comprises one transmitter coil and two receiver coils. These coils are implemented as copper traces, as depicted in Figure 4b. The first receiving trace is geometrically designed in a sinusoidal shape. The second coil is mechanically displaced by 90°, leading to a cosine-shaped feedback signal. The transmitter trace is excited by a high frequency sinusoidal current, typically between 2 MHz and 6 MHz [20], which is generated by the sensor chip on the PCB. This planar excitation coil builds an LC resonant circuit in combination with additional capacitors. Note that the inductance *L* is given by the coil trace itself. Electrical interferences can be reduced by ohmic resistors, where one pin is connected to the printed transmitter trace and the other pin is bonded to the capacitors, which are connected with the electrical Ground (GND). The LC resonant circuit generates a magnetic induction field and induces eddy currents in the rotating metallic target. According to the rotor position, a reaction field shields the excitation field in the area beneath the rotating target and induces voltages in the receiving traces. Consequently, inductive position sensors measure the disturbance of a magnetic field by a conductive target.

The sensor accuracy strongly depends on the selected coil design; hence, an ideal receiver coil geometry leads to an ideal sine- and cosine feedback voltage at the open circuit-designed coils. In operation, when the rotating target is facing the sensing area, the magnetic field induces the mentioned eddy currents into the target surface. As a result, these eddy currents produce a counter magnetic field, which leads to a reduction of the flux density beneath the target. This non-uniform flux density generates an electrical voltage at the receiver coil terminals according to Faraday's law of induction. The rotation of the target leads to a change in amplitude and polarity. A mathematical description of the inductive sensor principle is given by Equation (5). The variables are described as follows. The alternating field, which is created by the excitation, is given by $b_e(t)$. The alternating magnetic field, which opposes $b_e(t)$, is described by $b_s(t)$. The surface area of the receiving coil is represented by *A*.

$$VI = \frac{d\phi}{dt} = -\frac{d\int [b_e(t,x,y) + b_s(\mathrm{t},x,y)]dA}{dt} = \frac{d}{dt}\int b_e(t,x,y)dA + \int b_s(t,x,y)\,dA \qquad (5)$$

For signal processing, the feedback signals are amplified, rectified and filtered before being converted in the digital domain by an integrated ADC. Especially for this sensor type, the signal processing procedure is highly dependent on the application. Consequently, the sensor chip can

provide the absolute angular position of the rotor shaft in degrees, based on a trigonometric-based arctan function [21]. If this operation is performed in a subsequent module, e.g., a vehicle's Electronic Control Unit (ECU), only the demodulated feedback signals of the receiver coils are available, which leads to a reduction of signal processing complexity in the sensor, and therefore to lower costs [16,22].

In summary, inductive position sensors offer miscellaneous utilization for both automotive and non-automotive applications. The sensor's principle benefits are high accuracy, immunity against stray fields, an absolute rotor position determination from the electrical start up, and low costs due to both the low amount of copper wires and lack of magnetic material. The latter benefit makes the sensor more eligible for automotive applications compared to other sensor principles, such as magnetoresistive sensors, which will be described in the further context section. Another major advantage over other sensor technologies is that, in safety-critical applications, multiple sensing areas can be placed on the stator PCB. This enables it to combine multiple position sensors in one housing. In addition to the strengths mentioned, this sensor principle has some weaknesses, which are discussed below. Although the housing of the sensor requires only a small mounting area, a certain space is required for the PCB. In addition, the sensor is sensitive to AG variations in terms of mechanical misalignment or target eccentricity and wobbling.

Hereafter, a typical automotive application of an inductive position sensor system is presented. Since the inductive sensor is fully integrated on a single PCB, a small size can be realized, e.g., for the use in the electronic throttle valves of Internal Combustion Engines (ICEs), as shown in Figure 5.

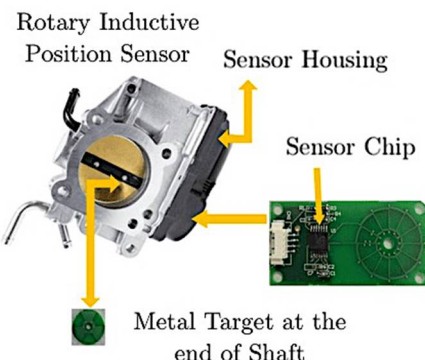

**Figure 5.** State-of-the-art inductive-based E-Gas actuator position sensor system. Reproduced from [23]. Copyright 1998-2020, Microchip Technology Inc.

In this example, the inductive position sensor acts as a feedback system for throttle valve control. The driver's request is delivered by the electronic accelerator pedal. The MCU determines the flap position and operates a Brushless Direct Current (BLDC) motor. To provide exact information about the flap position for the proper control of the actuator, an inductive position sensor is placed at the moving flap shaft.

Today, different electric motor manufacturers search for alternative technologies to replace the resolver position sensors with smaller and potentially more cost-efficient inductive sensor systems. Table 2 lists the strengths and weaknesses of the inductive position sensor principle.

**Table 2.** Summary of pros and cons of the inductive position sensor principle.

| | Attributes |
|---|---|
| Advantages | High accuracy and resolution |
| | Robust |
| | Immune to stray fields |
| | Requires no magnetic material |
| | Small installation space compared to the resolver technology |
| | Adaptable coil design for both linear and rotary movements |
| | Low system costs |
| | Redundant implementation possible |
| Disadvantages | Temperature-depending accuracy |
| | Thermal limitations of the PCB |
| | Sensitiveness to AG variations |

## 2.3. Hall Effect Position Sensor

The fundamental principle of this sensor is based on the eponymous effect discovered by Edwin Hall in 1879 [24] (Edwin Hall (1855–1938): American physicist and innovator of the Hall effect [25]). Hall effect sensors are widely utilized in various fields due to their main advantage of miniaturization and contactless measurement. Applications range from low-cost household products to complex industrial, automotive and aviation machines. Typical examples include BLDC motor commutation sensors, and crankshaft and level measurement. Figure 6 depicts a state-of-the-art Hall effect sensor.

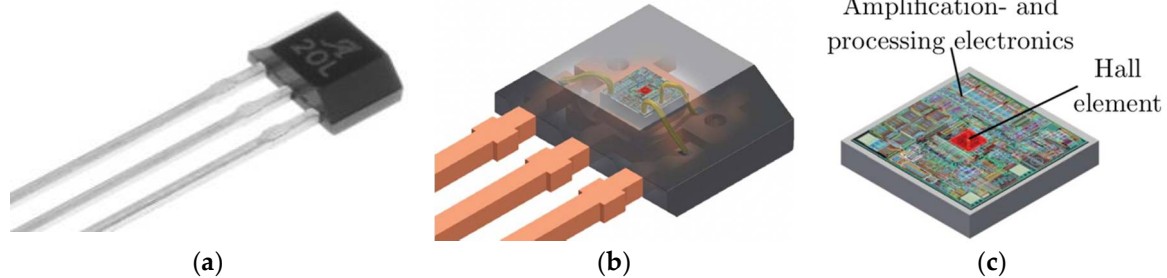

(**a**)　　　　　　　　　　　　(**b**)　　　　　　　　　　　　(**c**)

**Figure 6.** Hall effect sensor. (**a**) Electrical sensor component, reproduced from [26]. Copyright 2009-2013, Allegro MicroSystems, LLC; (**b**) Inner view of a Hall effect sensor comprising the mounted die and the bond wire connections to the external pins for supply, output signal and GND (from left to right) [27]; (**c**) Hall Integrated Circuit (IC) chip, reproduced from [27]. Copyright 2009–2013, Allegro MicroSystems, LLC.

The effect of the Hall sensor is based on the Lorentz-force, which acts on moving charge carriers. In this way, an electrical voltage is applied to the Hall plate, which results in an electrical current flow. When the current-carrying Hall plate is exposed perpendicularly to magnetic induction, e.g., by the use of a magnet, the charge carriers are deflected by a repulsion force to a certain angle, which is orthogonal to the applied field. This deviation is due to the aforementioned Lorentz-force. Accordingly, a Hall-voltage $V_H$, which is directed transverse to the current direction, can be picked-off between two opposite points on the Hall plate. The voltage is proportional to the magnetic field and the current. The Hall-voltage can be described as shown in Equation (6), where $I$ is the current, $B$ is the applied magnetic field, $q_0$ defines the charge of an electron, $N$ states the carrier density, $d$ represents the Hall plate thickness and $R_H$ specifies the Hall-coefficient [9,27]. The physical process is illustrated in Figure 7.

$$V_H \frac{IB}{q_0 N d} = R_H I \left(\frac{B}{d}\right) \tag{6}$$

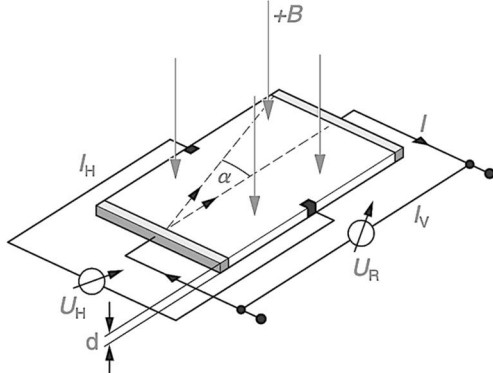

**Figure 7.** Illustration of the Hall effect principle, reproduced from [9]. Copyright 2007, Robert Bosch GmbH.

There are two main types of Hall effect sensors: linear devices and threshold types. Linear Hall sensors provide a proportional analog output voltage according to the magnetic field strength of the applied magnet. The output voltage moves into the supply direction (e.g., at a south pole) or GND (e.g., in case of a north pole). This can be achieved by a differential Operational Amplifier (OPAMP). As a result, the output voltage is dependent on the orientation of the magnet. When no magnet is detected at all, a bias voltage is present, which is also described as a Null voltage. Note that the fixed offset introduced is to avoid putting two power supplies in the processing circuit, which reduces the overall system costs [28]. A linear Hall sensor can be found, for example, in automotive throttle pedals.

The threshold type provides a digital state at the output (i.e., on and off) in case the field strength of the approached magnet obtains a certain magnetic amplitude or polarity. In terms of signal processing, this sensor configuration comprises a Schmitt-Trigger, which enables a solid, binary output voltage. In addition, these sensors can be configured as latching devices. Here, the sensor output is in the on-state when a south pole is detected (i.e., the magnetic operating point), and it turns off when a north pole appears (i.e., the magnetic release point). This characteristic is often used for rotary speed and position determination [28]. One example is the wheel speed sensor, which is used for an exemplary description of this sensor system in the following.

The measurement of the vehicle wheel speeds is i.a. used for safety and assistance systems, e.g., Anti-lock Braking System (ABS) and Electronic Stability Control (ESC). The design of wheel speed sensor systems is simple due to robustness. Since the Hall sensor is highly reliable, it can be placed close to the bearings of the wheels. A rotary ring includes permanent magnets for speed determination, as depicted in Figure 8.

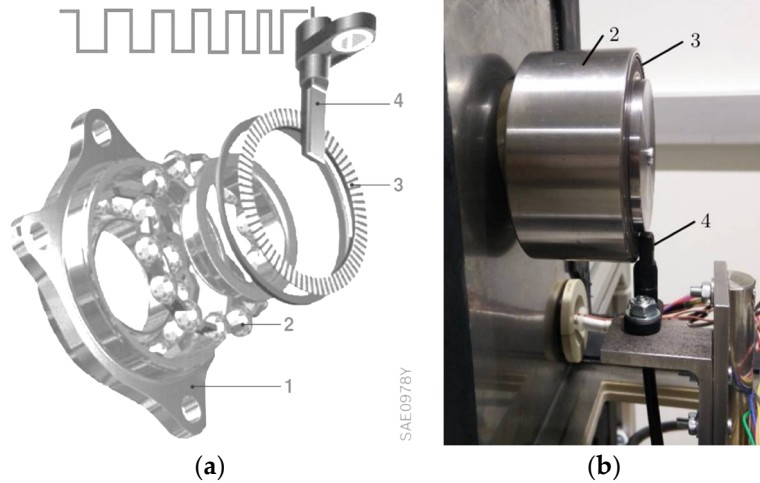

(**a**)　　　　　　　　　　　　　　　　　　　　　　　(**b**)

**Figure 8.** Hall-based wheel speed sensor. (1) wheel hub, (2) ball bearing, (3) permanent magnet ring, (4) wheel speed sensor. (**a**) Exploded assembly drawing, reproduced from [9]. Copyright 2007, Robert Bosch GmbH.; (**b**) Exemplary sensor on the sensor test bench.

Hall sensors provide a TTL signal to the vehicle ECU, which counts the pulses that correspond to the magnets passing the Hall element. When removing one magnet from the ring, a zero-mark can be obtained to enable absolute position detection.

In conclusion, Hall effect sensors are at their most versatile used when it comes to the measurement of position or rotary speed, since they can withstand rough environmental conditions (e.g., high temperature, shock, dust, contamination), and offer low system costs and miniaturization. However, Hall effect-based sensors are not found in powertrains, since a large amount of both Hall elements and magnets are required to achieve a sufficient accuracy, which increases the costs. Table 3 outlines the strengths and weaknesses of Hall sensors.

**Table 3.** Advantages and disadvantages of the Hall effect sensor technology.

|  | **Attributes** |
|---|---|
| Advantages | Reliable due to contactless measurement |
|  | Highly repeatable |
|  | Fast reaction |
|  | Broad temperature range |
|  | Low system costs |
| Disadvantages | External power supply required |
|  | Short range detection only in terms of AG |

*2.4. Magnetoresistive Position Sensor*

Magnetoresistive position sensors rely on a similar principle to the previously introduced Hall sensors, because they interact with an external magnetic field. A classification can be made in two types of magnetoresistive position sensors: the Anisotropic Magnetoresistive (AMR) type and the Giant Magnetoresistive type. The physical principle of both is based on Lorentz force [9,29]. A thin-film nickel-iron (NiFe) alloy coating, mostly the permalloy $Ni_{81}Fe_{19}$, with a thickness of 30 to 50 nanometers has an electromagnetically anisotropic characteristic. Accordingly, the material's electrical resistance changes depending on the direction of an external magnetization [9,29,30]. William Thomson discovered this effect in 1857, which describes electrons with different spin orientations possessing variable energy levels, resulting in a magnetic field-dependent conductivity when an external

magnetic field is present [29,30] (William Thomson (1824–1907): Scottish engineer, mathematician and physicist [31]). Equation (7) shows the electrical resistance as a function of the magnetization direction.

$$R(\theta) = R_0 + R\cos^2\theta = R_0\left(1 + \frac{\Delta R}{R}\right)\cos^2\theta \tag{7}$$

The term $\Delta R/R$ describes the maximum possible variation of resistance. The squared cosine term in Equation (7) depicts that the AMR sensor generates two sine-shaped curves over one full mechanical revolution. For the determination of the angular position, four magnet-depending resistors are usually combined into a Wheatstone bridge, where the second AMR resistor bridge is mechanically rotated by 45° on the substrate. This enables an additional signal, as a cosine function, to determine the angular position by an arctan computation. In other words, one bridge delivers a sine output signal, the other bridge a cosine signal. The calculation of the resulting voltage per bridge is given by Equation (8). Consequently, the position can be determined according to Equation (9).

$$V_{Bridge\ 1} = A(T)\sin(2\alpha);\ V_{Bridge\ 2} = A(T)\cos(2\alpha) \tag{8}$$

$$\theta = \frac{1}{2}\arctan\left(\frac{V_{Bridge\ 1}}{V_{Bridge\ 2}}\right) \tag{9}$$

Note that, in practical applications, both bridges are placed in one system by one manufacturing process to compensate for resistive temperature influence leading to a nearly equal amplitude per output signal. As a result, high-quality angular information without temperature influences (i.e., drifts) is achievable. Figure 9 demonstrates the principle and layout of an AMR angle sensor for a rotary measurement application in the range of 360°, as it is used in automotive applications.

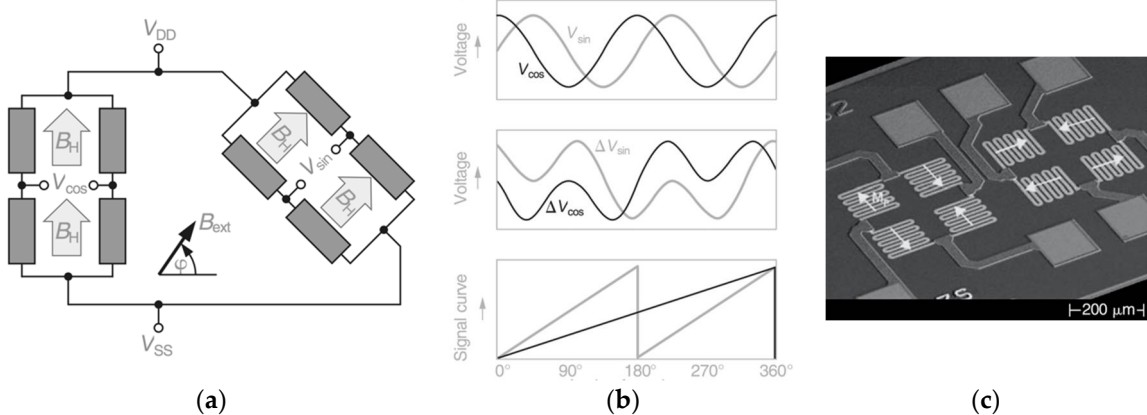

(a)　　　　　　　　　　　　　　　(b)　　　　　　　　　　　　　　　(c)

**Figure 9.** AMR sensor principle. (**a**) Sensor bridge circuit, where $B_H$ is the control induction, $U_{sin}$ and $U_{cos}$ are the measurement voltages, $U_{DD}$ and $U_{SS}$ are the supply voltages and $B_{ext}$ is the external magnetic field with the angle of rotation $\varphi$ of the magnet; (**b**) Resulting sensor signals; (**c**) Electro microscopical exposure of a 360° AMR sensor layout, reproduced from [9]. Copyright 2007, Robert Bosch GmbH.

AMR are typically used in BLDC-driven auxiliary units, e.g., in electric water/oil pumps or cooling fans. In case of the 360° variant, an additional planar coil is added, which is placed above the AMR resistors. When an electrical current is applied there, an auxiliary field is generated, which evokes changes in the output signals of each AMR bridge to differentiate the measurement range. With this sensor principle, a very high measurement accuracy of less than 1° can be achieved. However, the accuracy is strongly dependent on the utilized magnet and its central placement vertical to the sensing area.

The other sensor principle is based on the GMR effect, which was discovered in 1988. This sensing effect occurs in multilayer structures of interchangeable ferromagnetic and non-ferromagnetic materials.

In a simplified view, two alignment characteristics of this multilayer structure exist; the anti-parallel and the parallel alignment. In case of a lack of an applied external magnetic field, the structure is aligned in an anti-parallel order. This state is also called the initial state. When the multilayers are exposed to an external magnetic field (e.g., by a permanent magnet), the ferromagnetic layers are forced to align to the applied field direction. As a result of this alignment, the resistance of the multilayer structure decreases dependent on the applied magnetic field strength. The minimum resistance of the multilayers is reached when the applied magnetic field surpasses the anti-ferromagnetic coupling, so that all layers are oriented parallel in respect to the external field (saturation point). In conclusion, the GMR layers show a minimum resistance in parallel aligned magnetizations and a maximum resistance in counter-directed magnetization, as shown in Figure 10 [9,29,30,32,33].

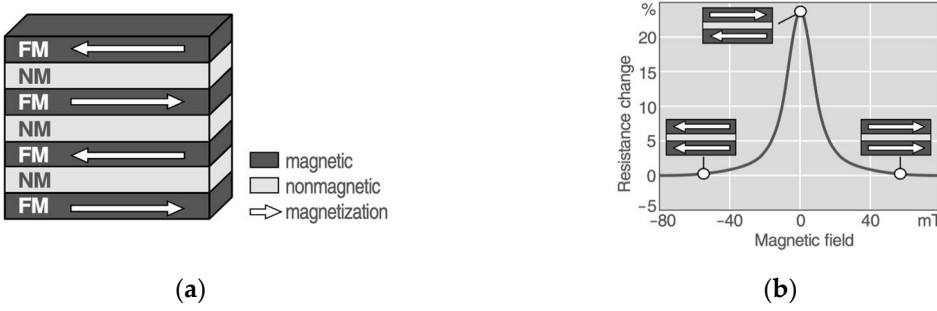

(**a**)                                                      (**b**)

**Figure 10.** GMR sensor principle. (**a**) GMR multilayer structure with FerroMagnetic (FM) and Non FerroMagnetic (NM) layers; (**b**) Resistance characteristic according to an applied magnetic field, reproduced from [9]. Copyright 2007, Robert Bosch GmbH.

The characteristic of a GMR multilayer sensor can be described according to Equation (10).

$$\text{GMR} = \frac{R_\text{min} - R_\text{max}}{R_\text{min}} = \frac{R_\text{saturated} - R_0}{R_\text{saturated}} \tag{10}$$

Since this effect is uniaxial, the resistance is proportional to the cosine of the angle $\theta$ of the applied external field according to Equation (11).

$$R = R_0 - \Delta R \cos(\theta) \tag{11}$$

Focusing on rotational GMR spin valve sensors for automotive applications, two Wheatstone bridges are used; the same as for the AMR principle to determine the position of the rotating shaft, as shown in Figure 11.

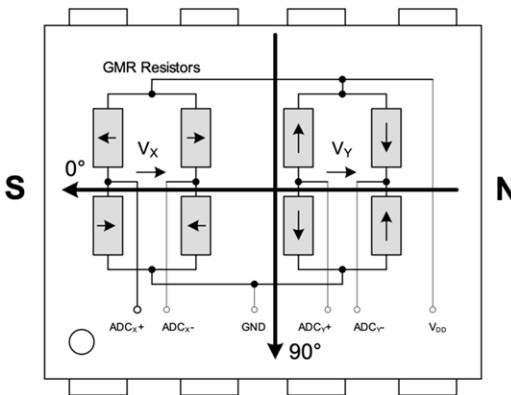

**Figure 11.** Exemplary GMR sensor die in a magnetic field (i.e., north pole (N) and south pole (S)) containing the sensitive GMR bridges, where $V_x$ and $V_y$ are the sinusoidal output voltages for further angle determination accessible via differential ADC outputs (i.e., ADC$_{x+}$, ADC$_{x-}$, ADC$_{y+}$, ADC$_{y-}$), reproduced from [34]. Copyright 2020, Infineon Technologies AG.

Considering one bridge for explanation, the output voltage of the bridge (i.e., $V_X$ and $V_Y$) resolves according to Equation (12) [35].

$$V_{X,Y} = \frac{\Delta R(1 - \alpha \Delta T)}{2R_0(1 + \alpha \Delta T) + \Delta R(1 - \alpha \Delta T)} \times V_{DD}; \ \Delta R = \frac{1}{2}\left(\frac{\Delta R}{R}\right)R_s \frac{iW}{d}\cos\left(\Theta_p - \Theta_f\right) \tag{12}$$

The parameters in Equation (12) are described as follows. The basic GMR and bridge resistance is given by $R_0$. $\Delta R$ is the variable resistance that depends on the direction of the magnetic field, and $V_{DD}$ represents the bridge biasing voltage. In addition, the sensor is temperature dependant. For that reason, $\alpha_0$ describes the absolute temperature coefficient of $R_0$, $\alpha_\Delta$ represents the absolute temperature coefficient of $\Delta R$ and $T$ the relative temperature to a reference temperature [35]. Owing to the fact that the resistance is dependent upon the GMR material, the width is described by $W$ and the thickness of the plate is given by $d$. The parameters $\Theta_p$ and $\Theta_f$ are the angles of magnetization of both the pinned and the free layers [36]. As a result, the orientation of the magnetic field—and thus the position of the magnet—can be determined by way of the resistance-dependent voltage. This analog voltage (i.e., $V_X$ and $V_Y$) is digitized by an ADC for further processing.

In contrast to the formerly introduced AMR position sensors, GMR sensors offer an inherent complete angular measurement range of 360°, where the calculated arctan position information repeats after exceeding 180°. Thus, no additional measures are required, contrary to the AMR sensor, where an additional planar coil is necessary. In addition, GMR sensors are very robust in terms of temperature, which is important for automotive applications (−40 °C to +150 °C). Even if the accuracy of GMR is in the same range as that of AMR, higher measurement signals, and magnets with lower field strength and therefore lower costs can be utilized. Unlike the AMR sensors, the GMR sensors do not only consist of one magnetic functional layer, but rather of a complex layer system. Additional descriptions regarding the differences of both introduced magnetoresistive sensor principles can be found in [29].

Concerning both types, AMR and GMR, the material of the rotary permanent magnet is vital to achieve a good angular measurement performance. Usually, magnets with a diametrically magnetized characteristic are selected. A scheme of the sensor setup is shown in Figure 12, where the permanent magnet is mounted at the end of a rotating shaft, the position of which is the information of interest. In general, the magnet needs to be placed very close to the sensing area of the AMR/GMR sensor, in an orthogonal way. As it can be seen, this design makes the sensor and therefore the accuracy vulnerable to mechanical deviations such as axial displacements and AG variations.

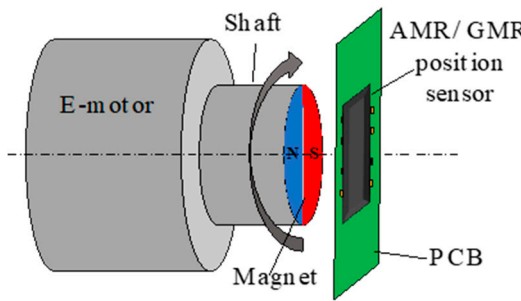

**Figure 12.** Typical setup of an AMR/GMR sensor.

Table 4 concludes both the strengths and weaknesses of the AMR/GMR position sensor technology.

**Table 4.** Listing of the strengths and weaknesses of magnetoresistive sensors.

| | Attributes |
|---|---|
| Advantages | Small installation space<br>High accuracy<br>Broad temperature range<br>Fast operation<br>High rotary speed range<br>Low system costs<br>Different data interfaces possible (e.g., analog, digital, bus) |
| Disadvantages | External power supply required<br>Short range detection only in terms of small AG<br>Sensitive to mechanical tolerances and eccentricity |

## 3. Experimental Characterization Method

In order to benchmark the performance of the previously introduced sensor technologies, an experimental method was applied by use of a rotor position sensor test bench. The corresponding sensor test bench has been specifically designed for rotor position sensor system error characterizations [12–14].

For analysis, the Device Under Test (DUT) is driven by an induction motor, which can reach a maximum speed of $n = 24,000$ rpm by a maximum angular acceleration of $a = 10,000$ rpm/s. This motor simulates the driving profiles of an electric powertrain. The speed is controlled by a highly accurate Hall-based position sensor with 8192 increments per rotation [37], which fulfills two tasks. Since the reference sensor provides an encoder emulation by the use of three TTL-based signals (i.e., A-, B- and Z-track), the speed is derived by the zero-position mark, which is the Z-track TTL signal for controlling the test bench drive comprising both the power electronics plus the control algorithm unit. In addition, the reference position information is used to evaluate the DUT angular accuracy in terms of potential angle error. The Hall-based reference sensor is doubly designed, and provides an angular resolution of $\theta < 0.1°$ at the maximum speed. Both angular data of the DUT position and the reference position are captured by a state-of-the-art 5 mega-samples per second (MS/s) Field Programmable Gate Array (FPGA)-based data acquisition unit [38]. This setup enables a universal analysis of various rotation position sensors with different interfaces (e.g., analog, digital, CAN-bus). In addition, a comprehensive sensor system characterization can be performed by integrating the corresponding DUT-ECUs [13].

The test bench not only provides high measurement accuracy, data acquisition rate and speed, but also automated adjustment of mechanical displacements of the DUT, as well as variation of sensor supply voltage and sensor operating temperature. For the investigation of the DUT sensitiveness regarding mechanical displacements (caused, for example, by production tolerances), the DUT stator is mounted on precisely displaceable motorized positioning actuators. This allows for the misalignment of the sensor stator relative to the rotor in four axes: $x$, $y$, $z$ and the vertical tilt. The variation of the

sensor supply voltage to simulate voltage fluctuations in the vehicle is performed by a controlled power supply. In addition, the ambient operation temperature can be varied by the use of a controlled climatic chamber surrounding the DUT during testing. Figure 13 shows the schematic test bench setup and a photo of the test facility.

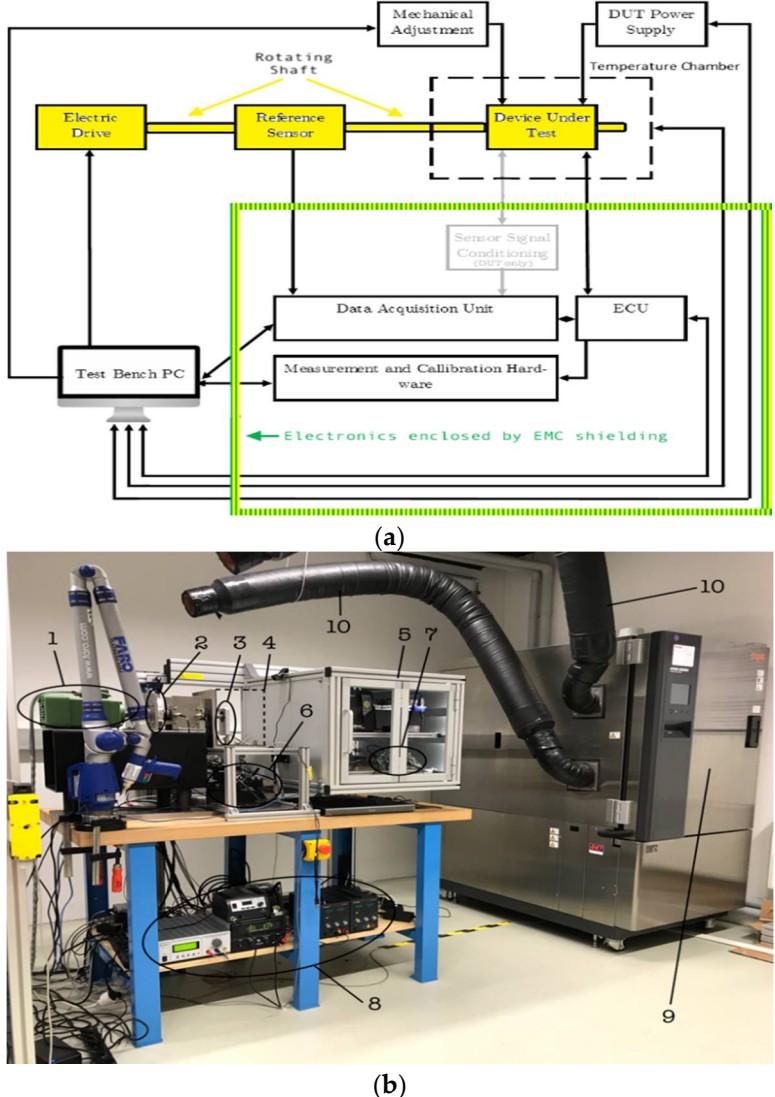

**(a)**

**(b)**

**Figure 13.** Sensor test bench (**a**) sensor test bench system overview; (**b**) sensor test bench and relevant components: 1, electric drive; 2, reference sensor; 3, device under test (DUT); 4, specific temperature chamber (removed for demonstration—see dashed box); 5, data acquisition housing (incl. electromagnetic compatibility (EMC) shielding); 6, motorized positioning actuators; 7, electronic control unit (ECU); 8, controlled power supply; 9, temperature conditioning system; 10, thermal tubes (for thermal transfer) [13].

In the present work, all investigated position sensors were evaluated in the same procedure. The resolvers are both powered and captured by a generic RDC evaluation board from Texas Instruments [39]. The angular information was provided by a TTL-based encoder emulation interface to the universal data acquisition card of the test bench to enable precise angle error determination. For the inductive sensors, the demodulated sine and cosine signals were captured by the analog inputs of the test bench data acquisition card. The angular position data of the magnetoresistive sensors were transferred by a CAN-bus interface onto the universal data acquisition card.

To enable both flexible and precise mounting of the different investigated sensor types, a universal adapter was used. The adapter system joins the rotary part of the DUT with the rotating drive shaft of the test bench by use of a high-precision clamping. To prevent vibrations at high speed testing, a specific balancing system was designed in the rotary adapter. Figure 14a depicts the universal sensor adapter with an exemplary rotating part of a resolver. Figure 14b shows the highly flexible stationary adapter, holding the stator of a resolver rotor position sensor as an example.

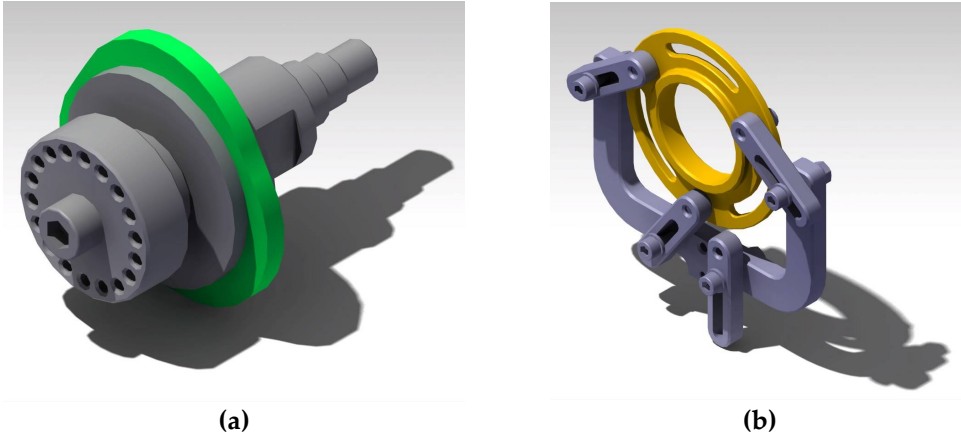

**(a)**          **(b)**

**Figure 14.** Rendering of the universal sensor adapter system. (**a**) Rotary part with an exemplary mounted resolver rotor (green part); (**b**) Stator adapter strap with an exemplary resolver stator, shown in yellow.

In the following section, the results of the comprehensive rotor position sensor investigations are presented. Figure 15 shows the definitions of reference coordinate system, tilt angle $\Delta\Phi$ and rotation speed $\Delta n$ used during the different test cases.

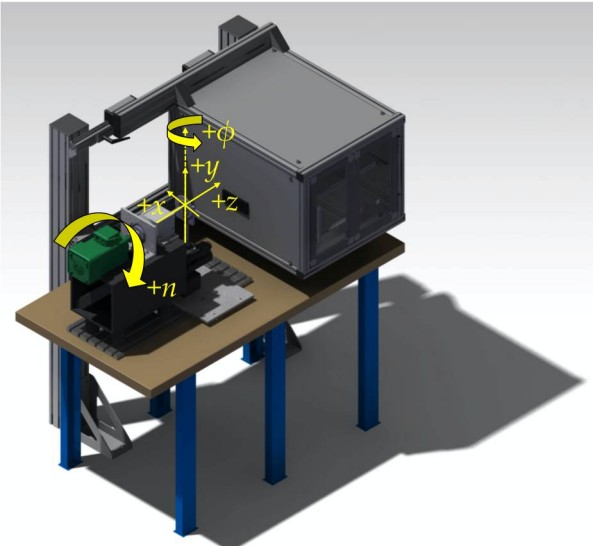

**Figure 15.** Coordinate system, tilt angle and rotational speed applied during the sensor system tests. The speed is defined as positive clockwise from $+z$ view and the tilt angle positive clockwise from bottom view ($+y$).

## 4. Results of a Comprehensive Rotor Position Sensor Technology Evaluation

The investigations focus on the angular accuracy of the previously introduced position sensor technologies (see Section 2). In total, 14 rotor position sensor types have been analyzed by the specified measurement series on the sensor test bench: six resolver-, seven inductive-/eddy current- and one AMR-position sensor. The considered sensors are from different manufacturers and differ in terms of their technical characteristics, e.g., number of pole pairs, mechanical sizes (i.e., rotor diameter) and the AG. Table 5 provides an overview of main data of the tested sensors. Observe that all sensors with an around shaft design are tested with an axial nominal AG of $z = 0$ mm, since the rotor is surrounded by the circular stator. However, a radial offset between stator and rotor still exists due to the rotor geometry.

**Table 5.** Technical details of the tested sensors.

| | Sample #1 | Sample #2 | Sample #3 | Sample #4 | Sample #5 | Sample #6 | Sample #7 |
|---|---|---|---|---|---|---|---|
| **Resolver Position Sensor** | | | | | | | |
| Number of Pole Pairs | 3 | 4 | 3 | 3 | 4 | 10 | |
| Maximal Operating Speed [rpm] | 17,550 | 15,000 | 17,550 | 22,000 | 18,000 | 9000 | |
| Operating Temperature Range [°C] | −40 to +150 | −40 to +180 | −40 to +150 | −40 to +150 | −40 to +150 | −40 to +150 | |
| Design | Around Shaft | Around Shaft | Around Shaft | Around Shaft | Around Shaft | Around Shaft | |
| Nominal AG [mm] | 0 | 0 | 0 | 0 | 0 | 0 | |
| **Inductive Position Sensor** | | | | | | | |
| Number of Pole Pairs | 3 | 3 | 3 | 2 | 4 | 4 | 2 |
| Maximal Operating Speed [rpm] | 24,000 | 24,000 | 24,000 | 20,000 | 20,000 | 20,000 | 20,000 |
| Operating Temperature Range [°C] | −40 to +150 | −40 to +150 | −40 to +150 | −40 to +150 | −40 to +150 | −40 to +150 | −40 to +150 |
| Design | Full Radial EoS | Around Shaft | Full Radial EoS | Arc 180° EoS | Arc 90° EoS | Arc 90° EoS | Arc 90° EoS |
| Nominal AG [mm] | 1 | 0 | 4.9 | 3 | 3 | 2 | 3 |
| **AMR** | | | | | | | |
| Maximal Operating Speed [rpm] | 7000 | | | | | | |
| Operating Temperature Range [°C] | −40 to +150 | | | | | | |
| Design | Full Radial | | | | | | |
| Nominal AG [mm] | 4 | | | | | | |

The evaluation results are based on a variety of test cases in terms of mechanical- and temperature-based variation, considering both varying sensor geometries and operational range. The sensor systems' error characteristics are compared in terms of the angular average peak-to-peak error. Since the peak-to-peak error is the worst-case scenario, the resulting value reveals, from the mean of the collected results, the angular peak-to-peak error of every single measured test case. Observe that the angular errors shown correspond to the mechanical error, as measured in relation to the drive shaft. The peak-to-peak error of every single measured test case $N$ is defined as

$$E_{\text{PP}}(N) = \max[\Delta\theta(t)] - \min[\Delta\theta(t)] \tag{13}$$

where $\Delta\theta(t)$ is the determined angular error:

$$\Delta\theta(t) = \theta_{\text{DUT}}(t) - \theta_{\text{Reference}}(t); \text{ with } \theta_{\text{DUT}}(t) = \arctan(V_{\text{sin}}(t)/V_{\text{cos}}(t)). \tag{14}$$

As a result, the utilized average peak-to-peak error can be determined according to (14).

$$E_{\text{PP, avg}} = \frac{1}{N} \int_0^N \left|E_{\text{PP}}(N)\right| \mathrm{d}N \tag{15}$$

### 4.1. Speed-Dependant Angular Error Evaluation

Figure 16 illustrates the angle error characterization of the investigated sensor technologies in the mechanical initial point. The initial point is defined as the point at which the sensor is centrally adjusted according to the data sheet. This implies that all mechanical variation parameters are nulled (i.e., $\Delta\phi = \Delta x = \Delta y = \Delta z = 0$ [°/mm]). Note that, for both the inductive sensors and the AMR sensor, the nominal AG, which was defined by the manufacturer (see Table 5), is applied. The measurements have been performed at a temperature of 25 °C, which is defined as Room Temperature (RT) in further context. The only parameter varied in this test case was the rotational speed of the drive axle according to the maximal speed specification of the manufacturer (e.g., $n = 17{,}550$ rpm, see Table 5-resolver sample #1). The minimum rotational speed, which is described in Figure 16 below, was chosen in a way such that a sufficient number of angular periods was available. The step size of the speed was chosen in hundreds of steps and, respectively, thousands of steps.

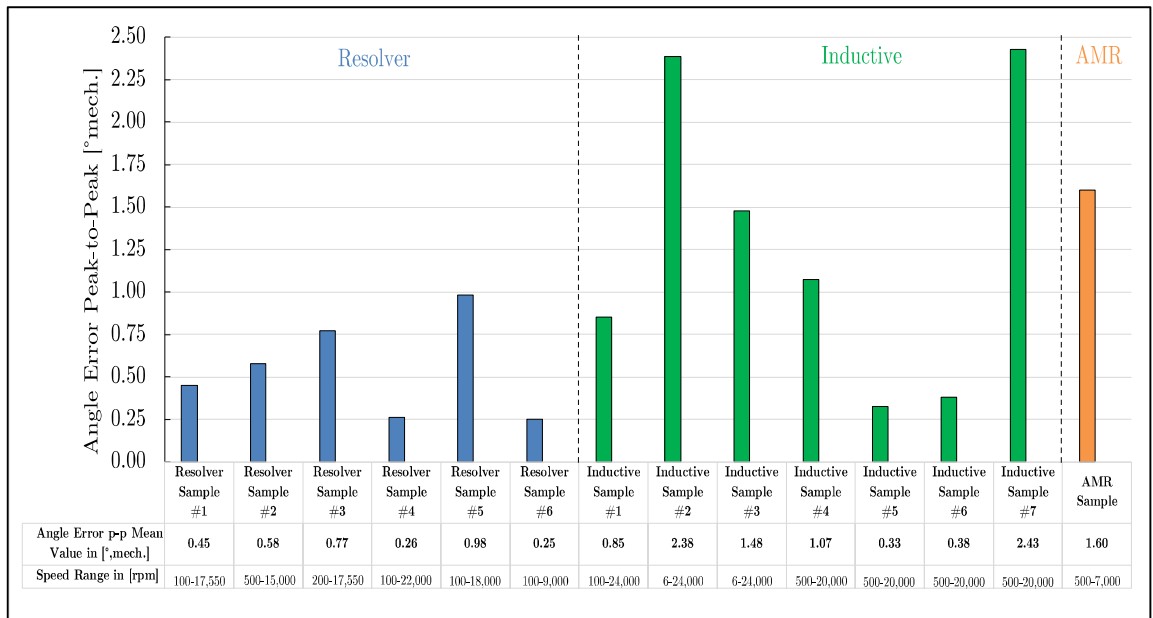

| | Resolver Sample #1 | Resolver Sample #2 | Resolver Sample #3 | Resolver Sample #4 | Resolver Sample #5 | Resolver Sample #6 | Inductive Sample #1 | Inductive Sample #2 | Inductive Sample #3 | Inductive Sample #4 | Inductive Sample #5 | Inductive Sample #6 | Inductive Sample #7 | AMR Sample |
|---|---|---|---|---|---|---|---|---|---|---|---|---|---|---|
| Angle Error p-p Mean Value in [°,mech.] | 0.45 | 0.58 | 0.77 | 0.26 | 0.98 | 0.25 | 0.85 | 2.38 | 1.48 | 1.07 | 0.33 | 0.38 | 2.43 | 1.60 |
| Speed Range in [rpm] | 100-17,550 | 500-15,000 | 200-17,550 | 100-22,000 | 100-18,000 | 100-9,000 | 100-24,000 | 6-24,000 | 6-24,000 | 500-20,000 | 500-20,000 | 500-20,000 | 500-20,000 | 500-7,000 |

**Figure 16.** Experimental comparison of the resolver-, inductive- and AMR-position sensor technology, based on a variable applied speed and both constant temperature ($T = 25$ °C $=$ RT) and constant centric sensor placement.

Considering the error characteristics of resolvers shown in Figure 16, it can be summarized that this technology offers a high accuracy. This is one of the reasons why this sensor type is the preferred choice in electric powertrain architectures today. Basically, it can be argued that a higher number of pole pairs results in better accuracy. However, this is not always the case, as a comparison of the different types shows (e.g., Resolver Sample #4 vs. Sample #5 and #6).

In general, inductive position sensors can reach a similar level of accuracy as resolvers (see Inductive Sample #1, #5, #6). Full radial sensor designs with a circumference of 360° deliver higher accuracy in principal, but other parameters, e.g., the AG, also have a considerable influence; compare Inductive Sample #1 and #3, with AG $= z = 1$ mm at sample #1 and AG $= z = 4.9$ mm at sample #3. Inductive Sample #2 with an around shaft design shows a lower accuracy, which might be caused by additional eddy currents induced by the steel-made drive shaft. Observing the arc designs of the inductive position sensors, the angular accuracy is comparably good to the resolvers and other inductive sensor designs. A resolver can be replaced by an inductive position sensor with two pole pairs and an arc design measuring a half circle (i.e., 180°); see Figure 16, Inductive Sample #4. The best accuracy is provided by two different 90° arc design inductive position sensors with four pole pairs. For example, a ten-pole pair resolver shows a similar accuracy to a four-pole pair arc 90° inductive sensor, while the inductive sensor has lower installation space requirements and, probably, lower costs. However, the number of pole pairs should be taken into account, since a lower amount of pole pairs (e.g., Inductive Sample #7) shows a comparatively high angle error compared to a sample with similar specifications and nominal AG (e.g., Inductive Sample #5).

Concerning the AMR technology, the advantage of miniaturization is diminished by the sensitivity of mechanical impacts. This can be seen in the comparatively low accuracy performance, which is in the middle range of the analyzed inductive sensors.

Besides a comparison of the different sensor technologies, Figure 16 shows that the sensor design, in terms of pole pairs and implementation (e.g., arc versus full circle in the case of the inductive sensor principle), strongly influences the accuracy. In the following subsection, mechanical displacements are also considered, in addition to the speed variation, in order to investigate the sensitivity of the sensors with regard to mechanical installation tolerances.

### 4.2. Multi-Mechanical Parameter Angular Error Characterization

The results of the multi-mechanical parameter variation tests, considering both geometrical misalignment and speed variations, are represented in Figure 17. The Figures A1–A3 in the Appendix A provide an overview of the conducted mechanical parameter variation range of each sensor, where the darker color represents the selected point. The angle error determination performed is based on Equations (13)–(15) and the average peak-to-peak error of each measurement series per parameter variation is considered.

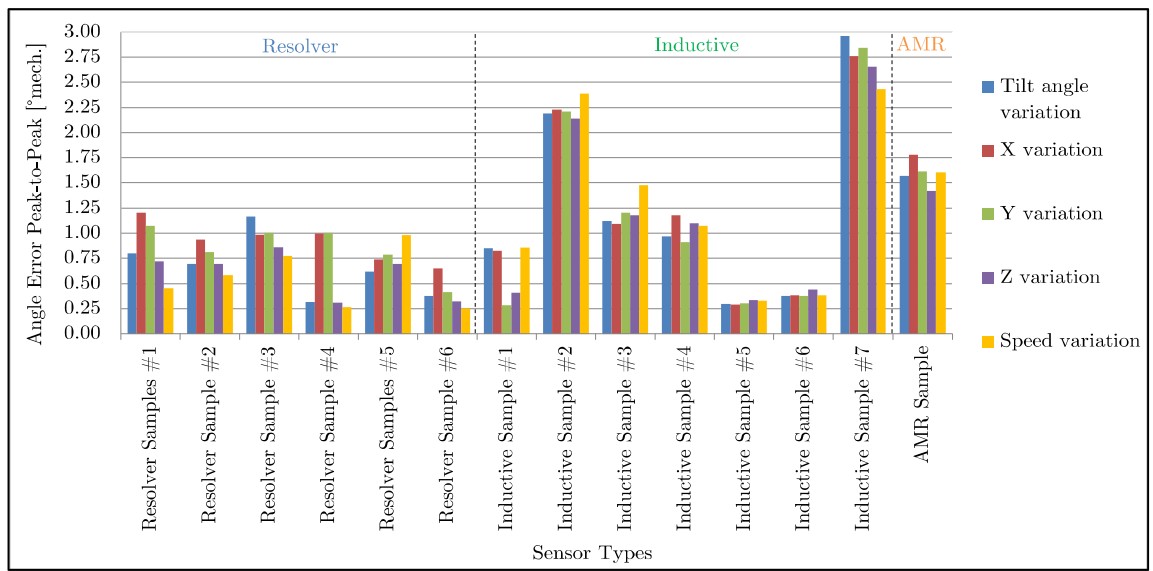

**Figure 17.** Benchmark of both mechanical and rotor speed parameter variations at RT = 25°, by use of the average angular peak-to-peak error from each test case, which is one parameter variation.

The inductive sensor principle is capable of reaching the characteristics of the resolver technology in terms of mechanical deviations and speed variation (e.g., Inductive Sample #5 & #6). The design of the inductive position sensor is highly responsible for the angular accuracy, e.g., arc design vs. 360° design.

The magnetoresistive sensor principle is suitable for miniaturized applications, and the angular accuracy performance is in the range of the other two sensor types. Furthermore, the robustness in terms of mechanical misalignment in all axes shows that some sensors offer a very good performance (e.g., the Resolver Sample #3), while other sensors are more sensitive to mechanical deviations (e.g., Inductive Sample #1). In general, it can be concluded that the greater the difference between the respective bars of a specific sensor in the figure above, the more sensitive it is to *x*-, *y*-, *z*- and tilt-orientation-related tolerances.

### 4.3. Temperature Variation-Based Angular Accuracy of Non-Resolver Position Sensors

The error response at different operating ambient temperature conditions was investigated over both the speed range and the mechanical initial position. Since it is known that resolver position sensors are very robust against temperature influences [40], only the inductive- and AMR-based sensors were examined here. Different thermal conditions can lead to angle error distortion for these sensor types, owing to the temperature coefficient of the PCB material (i.e., the inductive sensor) or the magnetic material (i.e., the magnetoresistive position sensor). Figure 18 depicts the error characteristics based on the thermal influences, which were provoked on the sensor system test bench by use of the climatic conditioning system. The angular error for each applied speed at a certain temperature was evaluated based on Equations (13)–(15).

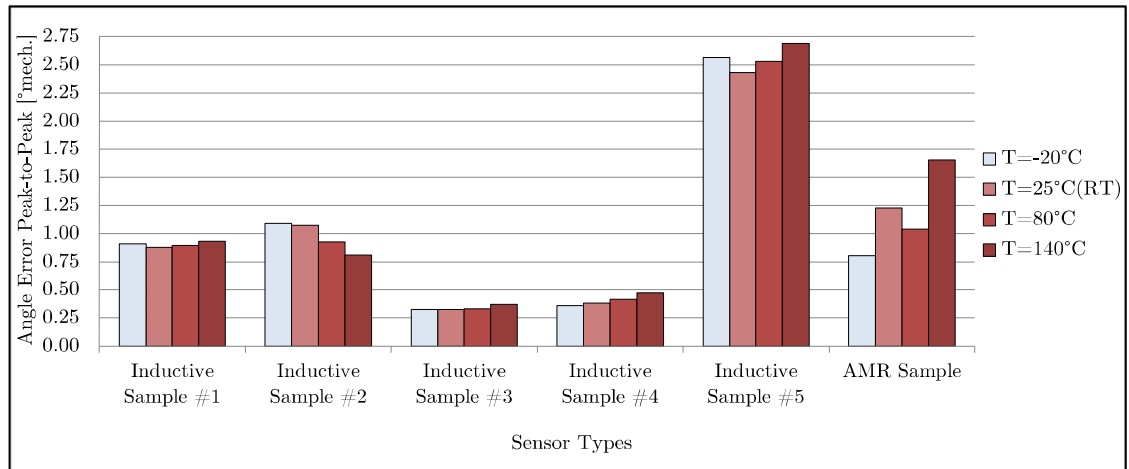

**Figure 18.** Inductive- and AMR-based sensor accuracy characteristics in the sensor systems' sweet spots at different operational ambient temperatures.

Besides the fact that the sensor systems' accuracy strongly depends on the design of both the target (i.e., the rotor) and the stator coil structure, temperature plays an important role for accurate operation. Aside from Inductive Sample #2, with its around shaft design, all the other evaluated devices show the highest angular error at maximum temperature. It can also be seen that thermal changes do not influence the accuracy very significantly when considering the inductive sensors. However, the AMR sensor principle, which is based on the magnetoresistive effect, shows the best performance at the lowest temperature and worst performance at maximum temperature. The selected temperature of T = 80 °C is a common operational condition in drivetrain applications, and T = 140 °C represents high load conditions, which typically occur for a short duration. It can be concluded that the angular accuracy varies in the range of a maximum of 25% for inductive sensors and about 120% for the AMR. Either way, the absolute angular deviation is about 0.25° for inductive sensors and about 0.90° for the AMR.

### 4.4. Validation of the Presented Results

This subsection provides a validation of the previously presented results. To summarize, the experimentally-based evaluation method consisted of three main phases. In the first phase, the DUT was mounted on the test bench on both the motorized stator positioning system (Figure 13) and the rotating shaft (Figure 14). In addition, the electrical installation was conducted, i.e., the sine and cosine feedback signals were connected with the data acquisition unit of the test bench. The experimental characterization was performed in phase two of the investigation process, which started with loading the test cases that are defined for each parameter variation (e.g., speed $n$, temperature $T$, other mechanical parameters $x$, $y$, $z$, $\Phi$). In the following, the test procedure was started by conditioning the influencing parameters on the test bench. The process of the determination of the angular information of both the reference sensor and the DUT was repeated for every test case. This process included signal acquisition and data storage. After completing every test case, the data was processed in phase three. This comprised the conversion of the mechanical reference angle to the electrical angle value by consideration of the number of pole pairs of the DUT. Furthermore, both electrical angles were compensated in terms of the angular offset. As a result, each position started from 0°. This process was repeated after every overflow, i.e., exceeding 360°. Based on these data, the electrical angle error was subsequently determined, including both the computation and storage of the angular peak-to-peak error for each test case number. To achieve the final result for each parameter variation, which is presented in the figures above, the mean peak-to-peak error of every single parameter variation was conducted. The previously described process is graphically summarized in Figure 19.

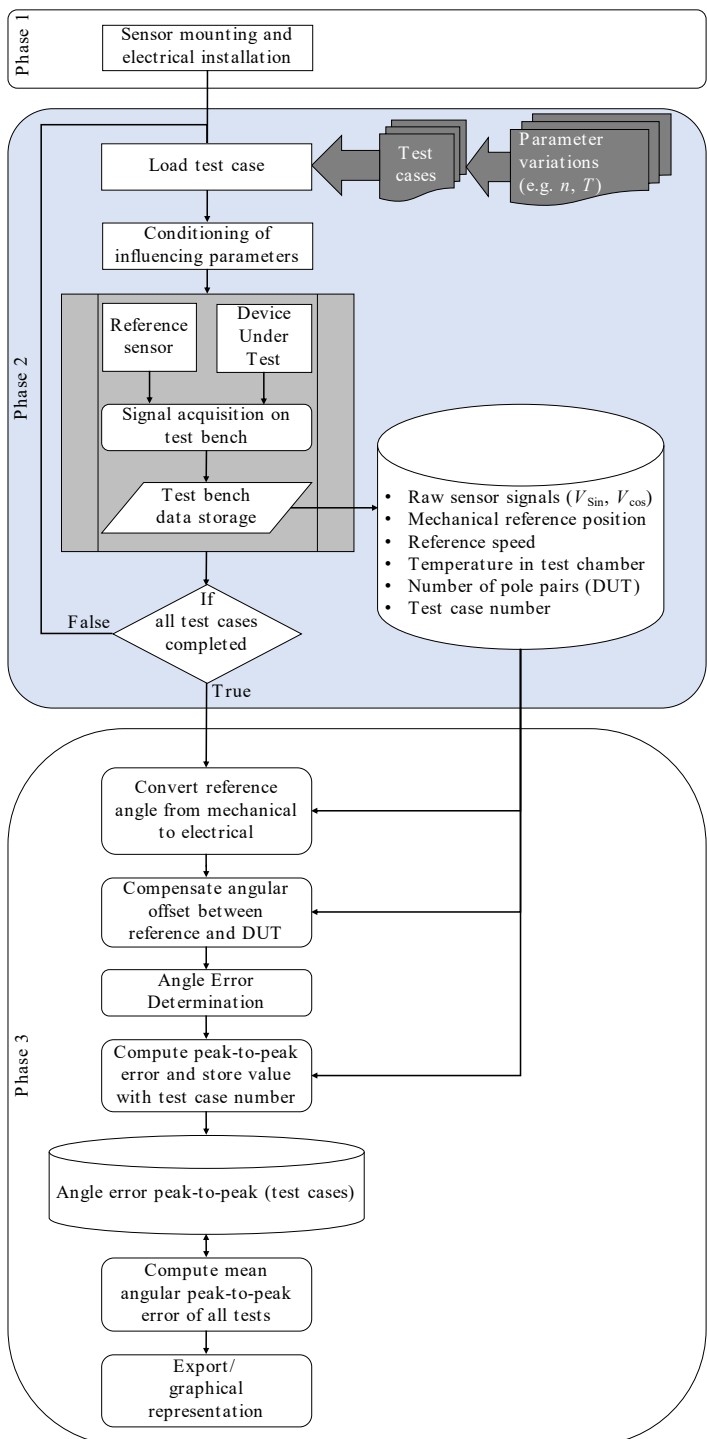

**Figure 19.** Illustration of the sensor characterization process.

Sensor signal data processing and validation are exemplarily shown for the Inductive Sample #4 at a constant speed of $n$ = 4000 rpm under RT, which includes both the speed variation and temperature variation test. The resulting signals can be seen in Figure 20.

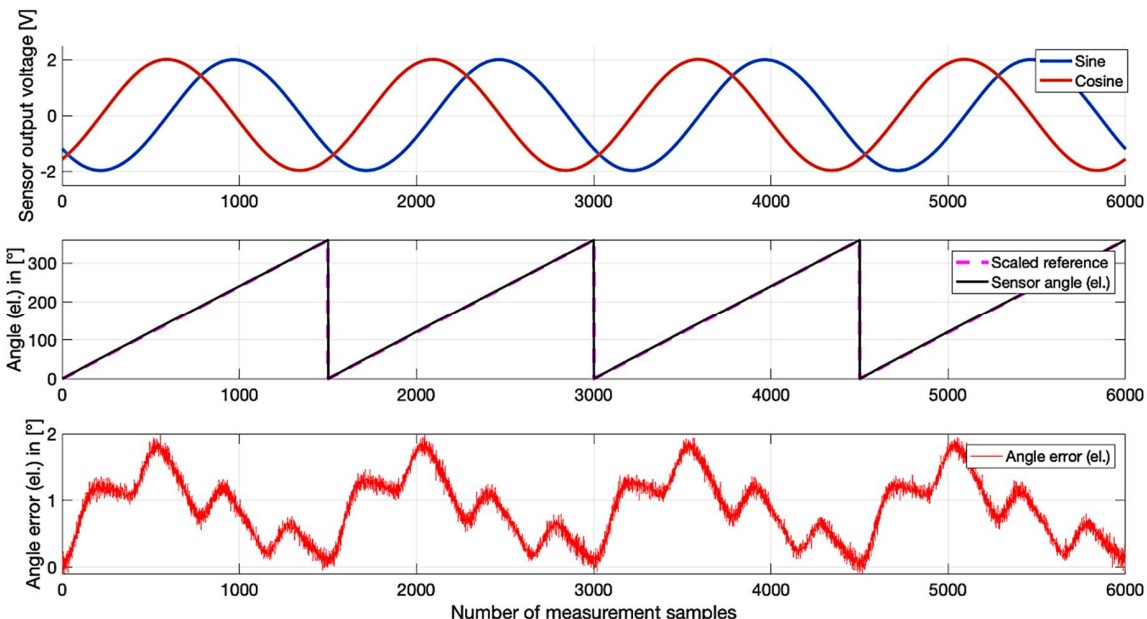

**Figure 20.** Measurement and processed signals of the Inductive Sample #4 at *n* = 4000 rpm and RT.

The sensor delivers two sinusoidal signals, i.e., sine and cosine (first sublot, Figure 20). The resulting electrical position of the sensor is illustrated in the second subplot of Figure 20. Based on the characterization method, the scaled and offset compensated reference angle can also be found in this graph. The resulting angle error of four periods, i.e., twice the number of pole pairs, is shown in the third subplot in Figure 20. The periodical error can be seen at a glance. Observe that the evaluation was performed over the recorded samples. Consequently, the peak-to-peak error is 2.11° electrically and 1.06° mechanically, since the pole pairs of this sample are two. Table 6 outlines the results for all measurements of the mentioned parameter variation, including the exemplary test, which is presented above. In addition, the mean peak-to-peak error for this specific test series is stated, as considered in Figure 16.

**Table 6.** Results of the speed variation test of Inductive Sample #4.

| Speed *n* [rpm] | Temperature *T* [°C] | Angle Error Peak-to-peak [°, el.] | Angle Error Peak-to-peak [°, mech.] |
|---|---|---|---|
| 1000 | 25 | 2.09 | 1.05 |
| 2000 | 25 | 2.02 | 1.01 |
| 4000 | 25 | 2.11 | 1.06 |
| 6000 | 25 | 2.03 | 1.01 |
| 8000 | 25 | 2.04 | 1.02 |
| 10,000 | 25 | 2.27 | 1.13 |
| 20,000 | 25 | 2.57 | 1.29 |
| - | - | Mean peak-to-peak error [°, mech.] | 1.08 |

## 5. Discussion

The present research provides a comprehensive investigation of the angular accuracy of different rotor position sensor technologies in terms of a variation of multi-mechanical and thermal parameters. The considered sensor types in this work include resolver-, inductive-/eddy current-, Hall- and magnetoresistive-based position sensors.

The results show that the inductive sensors are able to reach the performance of resolver position sensors. It was found that the accuracy of the inductive sensor technology strongly depends on

the sensor design. As an example, a full circle design has the potential to perform in a comparable accuracy-range to a resolver; the arc design can even achieve equal or better accuracy characteristics. It has to be considered that both parameters, the number of pole pairs and the nominal AG, are important influencing factors. The AMR sensor principle comes with a great advantage for miniaturized applications, but its angular accuracy is heavily reliant on mechanical tolerances.

When operating the position sensor in a powertrain, the accuracy in terms of thermal deviations is an important criterion. Since resolver sensors show nearly constant accuracy over the holistic temperature range [38], non-resolver-based position sensors were investigated according to their angular accuracy under different automotive ambient temperatures. The tested inductive- and magnetoresistive- position sensors show higher sensitivity to temperature changes, because the PCB material is more sensitive in terms of geometrical form changes at induction sensors, and the magnetic performance varies as a function of the ambient temperature in the case of magnetoresistive sensors. In general, the accuracy of the sensor deteriorates at higher temperatures.

Figure 21 shows the potentials of the different sensor technologies under consideration of selected performance indicators, which is relevant for the application in automotive powertrain systems. Based on the results of the study, it can be stated that the resolver is one of the most accurate and, at the same time, robust sensors for powertrain applications. However, complex signal processing (leading to relatively high costs), and comparatively high weight and mechanical size are disadvantages of this sensing principle.

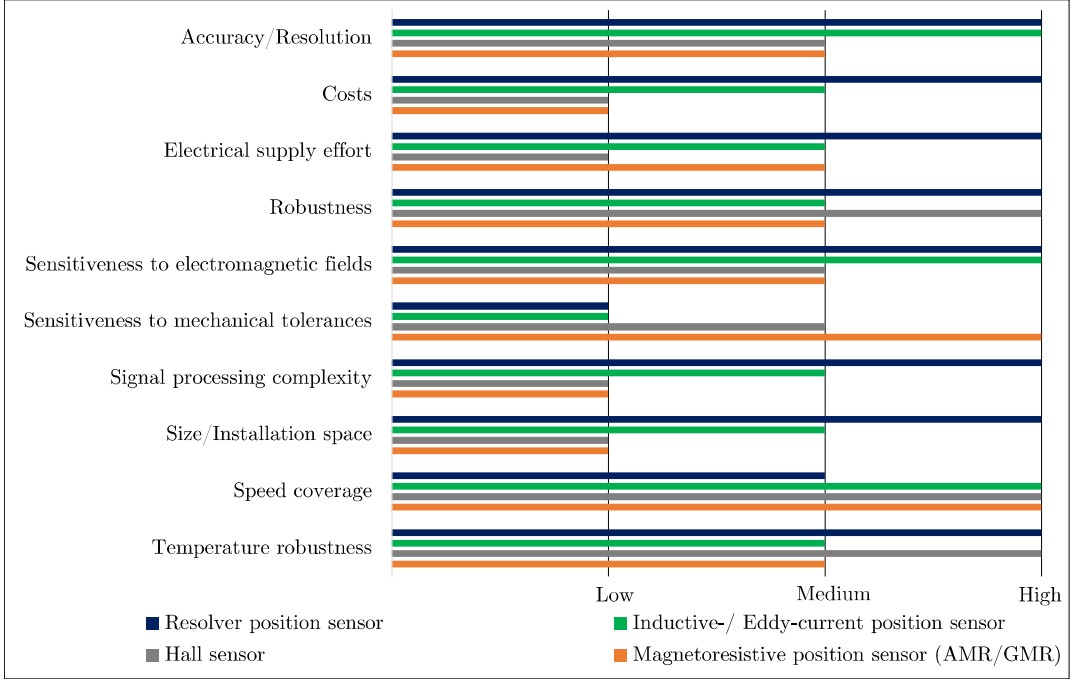

**Figure 21.** Comparison of the different sensor technologies related to selected performance indicators.

Inductive sensors are capable of fulfilling key requirements to a high level, comparable to resolver performance. An important characteristic is the design (i.e., full 360° vs. arc design) of the sensor, since it has an influence on the accuracy. Furthermore, the thermal sensitivity of the PCB carrier material influences the angular error in the temperature-sensitive application of traction motors. For AMR and GMR sensors, current state-of-the-art semiconductor sensor manufacturing technology enables a very precise determination of the measured rotating shaft. Inductive- and eddy-current position sensors offer less rotor mass and a smaller sensor housing. However, the present research deals with increasing the sensor performance to achieve equal fidelity compared to the commonly-used resolver technology, which is accomplishable by improving the coil design. Hall sensors can be used in large

numbers to measure the rotor position quite precisely (e.g., as a reference sensor utilized on the test bench). When applying this sensing method, the larger magnet size needs to be taken into account compared to magnetoresistive sensors. Larger magnets including a higher number of Hall elements (required for high accuracy) also increase the housing size, which might restrict the usability for automotive applications with high demands for compact design. The investigated AMR/GMR sensors offer higher resolution and, at the same time, a small installation space. Therefore, when a precise position determination of the rotating shaft is demanded, e.g., at small BLDC motors, magnetoresistive sensors are preferable. In the case of low requirements on the position information, reduced numbers of Hall elements, e.g., three elements mechanically displaced by 120°, might be sufficient.

Evaluating the sensor costs, the resolver is the most high-priced system. This is due to the fact that a high amount of copper wires and a complex winding process is required, similar to an electric motor. Also, signal processing is effortful, demanding a performant microcontroller (μC), raising the system costs. Inductive position sensors are cheaper compared to resolvers, since all sensing components can be placed on a low-cost PCB. Only the rotor needs to be made of a ferromagnetic material. Especially in mass production, the inductive sensor principle can play to its benefit in comparison to the resolver. Hall-based and magnetoresistive sensors can also be produced relatively cheaply; hence, semiconductor mass production allows low cost manufacturing. The only matter of expense in these sensor systems is the required magnet(s), the costs of which depend mainly on the magnet size and material.

The electrical supply of the sensor plays a key role when it comes to efficiency in automotive applications. Figure 21 shows that the resolver has the highest electric power demand because of its relatively high excitation currents and voltage. Thus, a powerful driver stage is inevitable, increasing the costs and influencing other electrical components in terms of Electromagnetic Compatibility (EMC). Inductive and magnetoresistive sensors do not need such high power for sensor operation. The Hall-effect sensor only requires low constant DC-Voltage to detect a magnet and change the output stage.

Resolvers are used in various powertrains of BEVs and HEVs offering robust rotary measurements today. Even in aviation and military practice, this sensor principle can be found because of its high durability and fail safety behavior. The same characteristics are applicable to the Hall sensor. Regarding inductive and magnetoresistive sensor principles, the overall robustness can be considered equal in case that a solid housing is designed. There are rough conditions for accurate position sensing in electric powertrains, especially when the sensor is mounted close to the electric machine, which produces strong electromagnetic fields. Resolver and inductive position sensors are more sensitive to electromagnetic influences than Hall- and magnetoresistive sensors. Both resolvers and inductive sensors rely on a similar principle, using a ferromagnetic rotor material to determine the position of the rotating shaft. An external field can influence the feedback signals and so distort the angular information. Today, different research activities are ongoing for inductive position sensors to improve their performance in terms of EMC. All magnet-based rotor position sensors show a certain sensitivity against stray fields. For that reason, they are rated as "medium", compared to resolver and inductive sensor technology.

Sensitiveness to mechanical tolerances is one of the investigations in this work, so this attribute is rated based on each sensor working principle. Mechanical tolerances can occur during installation processes, resulting in mechanical offset from the ideal-considered centric position, leading to an angular measurement error. For resolvers and inductive position sensors, this sensitivity is quite low. Resolvers usually operate under certain mechanical boundaries (i.e., the inner side of the stator), and are mainly sensitive to vertical tilt and axial offsets. Inductive position sensors show quite similar good characteristics, because the coil design can be optimized to the operating point. However, mechanical deviations of the rotating target in respect to the stator can cause biased angular information. Hall elements sense only in a relatively small distance. Larger AG ranges between sensor element and magnet or fluctuations (e.g., wobble) distort the measurement result. AMR/GMR position sensors are

typical EoS applications, and show high sensitivity regarding mechanical tolerances. Minor axial- or radial displacements cause relatively high angular errors.

In terms of signal processing complexity, the resolver technology requires more complex signal processing, comprising digitalization, filtering, demodulation of the AM feedback signals, trigonometric-based angular calculation and optional observer algorithms to process the rotor position. All these steps, performed in both hardware and software, usually on a multi-core µC, are integrated in a so-called Resolver-To-Digital Converter (RDC), also denoted as an R2D. The high complexity of signal processing is cost-intensive, and can cause additional angular errors, which can be seen as potential drawback of this technology. In inductive position sensors, the feedback signals on the sine- and cosine traces are also AM signals, which need to be demodulated. Typical sensors using this principle offer signal processing as a powerful one-chip solution directly on the PCB. In this way, the resulting angular information can be read in directly by the MCU without drawing computational power, which reduces the effort of angle determination compared to resolver technology. Hall sensors and magnetoresistive sensors are less complex in terms of signal processing effort. Hall sensors provide either an analog or a digital output signal to the subsequent control unit (e.g., ECU, MCU), where, in the case of an analog signal, the information only needs to be digitalized and interpreted. For most applications, digital signals are used in form of TTL pulses. The position is then calculated by counting the pulses, enabling a simple signal processing complexity. Magnetoresistive sensors contain Wheatstone Bridges, delivering (magnet) position dependant sinusoidal voltages at the output as a result of resistance changes. These electrical circuits are relatively low in complexity, and are designed similarly to analog Hall sensors. In this case, the analog voltage is interpreted by an external controlling unit to determine the absolute rotary position.

Resolvers tend to be relatively large radial sensors with considerable weight and rotor inertia. Inductive position sensors only require space for the PCB-based stator and a thin rotary target with distinctly less inertia compared to the resolver. Hall and magnetoresistive sensors are very small components with a chip size of a few millimeters. However, in the case of Hall-sensors, a higher number of elements might be necessary to provide the specified accuracy. Miniaturization can be enabled due to semiconductor manufacturing; consequently, these sensor types need less installation space than resolvers and inductive position sensors. One parameter to be considered is the size of the magnet that is used. All sensors, apart from resolvers, can operate at very high speeds ($n \geq 20$ k rpm), due to the relatively small rotor inertia.

The findings of the temperature-related investigations show that resolvers cover the full automotive range due to their thermally robust sensor design. Here, temperature-biased errors stem only from the thermal resistance change of the copper windings. Hall sensors can also withstand high and low temperatures. AMR/GMR sensors can be designed to be temperature-compensating to reduce temperature-biased angular errors. For both Hall and magnetoresistive sensors, the only limiting criterion is the Curie Temperature $T_C$ of the magnet. Inductive position sensors are able to operate within the automotive temperature range, but the thermal properties of the PCB substrates need to be taken into account.

## 6. Conclusions

The present research provides a detailed benchmark of modern rotor position sensor technologies for application in automotive electric drive trains, considering different influencing factors. The investigated sensor systems were evaluated considering the typical performance indicators of automotive powertrain applications, including sensor accuracy, sensitiveness to mechanical tolerances and electromagnetic fields, effort of signal processing and temperature robustness, as well as the required installation space and cost aspects. For this purpose, a specific sensor test bench, including sophisticated measurement equipment, mechanical and electrical misalignment equipment, as well as a thermal conditioning system, was used. The measurement results are displayed in diagrams and discussed to elaborate the strengths and weaknesses of the different investigated sensor technologies.

For application in modern electric powertrain systems, resolver (as the most used technology) and inductive sensors show the best potentials due to their high accuracy and robustness against mechanical, thermal and electrical disturbances. For non-traction applications, e.g., auxiliaries driven by BLDC-motors, magnetoresistive and Hall-based sensors seem to be more suitable, due to lower cost and installation space. Nevertheless, the position sensor technology has to be selected according to the requirements of the specific application. In this context, the present work contributes to the selection of suitable sensor technology in the development of new electric propulsion systems.

**Author Contributions:** Conceptualization, C.D. and M.H.; methodology, C.D.; software, C.D.; validation, C.D.; formal analysis, C.D.; investigations, C.D. and M.H.; resources, C.D. and M.H.; data curation, C.D.; writing—original draft preparation, C.D.; writing—review, M.H.; visualization, C.D.; project supervision, M.H. All authors have read and agreed to the published version of the manuscript.

**Funding:** Open access funding provided by Graz University of Technology.

**Conflicts of Interest:** The authors declare no conflict of interest.

## Appendix A

| Measurement points [°] | Tilt Variation | | | | | | | | | | | | | |
|---|---|---|---|---|---|---|---|---|---|---|---|---|---|---|
| | #1 & #2 | #3 | #4 | #5 | #6 & #7 | #8 | #1 | #2 | #3 | #4 | #5 | #6 | #7 | AMR Sample |
| -2 | | | | | | | | | | | | | | |
| -1.5 | | | | | | | | | | | | | | |
| -1 | | | | | | | | | | | | | | |
| -0.75 | | | | | | | | | | | | | | |
| -0.5 | | | | | | | | | | | | | | |
| -0.4 | | | | | | | | | | | | | | |
| -0.35 | | | | | | | | | | | | | | |
| -0.3 | | | | | | | | | | | | | | |
| -0.25 | | | | | | | | | | | | | | |
| -0.2 | | | | | | | | | | | | | | |
| -0.1 | | | | | | | | | | | | | | |
| 0 | | | | | | | | | | | | | | |
| 0.1 | | | | | | | | | | | | | | |
| 0.2 | | | | | | | | | | | | | | |
| 0.25 | | | | | | | | | | | | | | |
| 0.3 | | | | | | | | | | | | | | |
| 0.35 | | | | | | | | | | | | | | |
| 0.4 | | | | | | | | | | | | | | |
| 0.5 | | | | | | | | | | | | | | |
| 0.75 | | | | | | | | | | | | | | |
| 1 | | | | | | | | | | | | | | |
| 1.5 | | | | | | | | | | | | | | |
| 2 | | | | | | | | | | | | | | |

**Figure A1.** Tilt variation measurement plan for all evaluated sensors; blue, resolvers; green, inductive sensors; orange, AMR sensor.

| Measurement points [mm] | x- and y- Variation | | | | | | | | | | | | | |
|---|---|---|---|---|---|---|---|---|---|---|---|---|---|---|
| | #1 & #2 | #3 | #4 | #5 | #6 & #7 | #8 | #1 | #2 | #3 | #4 | #5 | #6 | #7 | AMR Sample |
| -1.2 | | | | | | | | | | | | | | |
| -1 | | | | | | | | | | | | | | |
| -0.8 | | | | | | | | | | | | | | |
| -0.75 | | | | | | | | | | | | | | |
| -0.6 | | | | | | | | | | | | | | |
| -0.5 | | | | | | | | | | | | | | |
| -0.4 | | | | | | | | | | | | | | |
| -0.35 | | | | | | | | | | | | | | |
| -0.3 | | | | | | | | | | | | | | |
| -0.25 | | | | | | | | | | | | | | |
| -0.2 | | | | | | | | | | | | | | |
| -0.15 | | | | | | | | | | | | | | |
| -0.1 | | | | | | | | | | | | | | |
| 0 | | | | | | | | | | | | | | |
| 0.1 | | | | | | | | | | | | | | |
| 0.15 | | | | | | | | | | | | | | |
| 0.2 | | | | | | | | | | | | | | |
| 0.25 | | | | | | | | | | | | | | |
| 0.3 | | | | | | | | | | | | | | |
| 0.35 | | | | | | | | | | | | | | |
| 0.4 | | | | | | | | | | | | | | |
| 0.5 | | | | | | | | | | | | | | |
| 0.6 | | | | | | | | | | | | | | |
| 0.75 | | | | | | | | | | | | | | |
| 0.8 | | | | | | | | | | | | | | |
| 1 | | | | | | | | | | | | | | |
| 1.2 | | | | | | | | | | | | | | |

**Figure A2.** Test case plan for the *x* and *y* variation to evaluate all available sensors; blue, resolvers; green, inductive sensors; orange, AMR sensor.

| Measurement points [mm] | z- Variation | | | | | | | | | | | | | |
|---|---|---|---|---|---|---|---|---|---|---|---|---|---|---|
| | #1 & #2 | #3 | #4 | #5 | #6 & #7 | #8 | #1 | #2 | #3 | #4 | #5 | #6 | #7 | AMR Sample |
| -2 | | | | | | | | | | | | | | |
| -1.6 | | | | | | | | | | | | | | |
| -1.5 | | | | | | | | | | | | | | |
| -1.4 | | | | | | | | | | | | | | |
| -1.2 | | | | | | | | | | | | | | |
| -1 | | | | | | | | | | | | | | |
| -0.832 | | | | | | | | | | | | | | |
| -0.8 | | | | | | | | | | | | | | |
| -0.7 | | | | | | | | | | | | | | |
| -0.6 | | | | | | | | | | | | | | |
| -0.5 | | | | | | | | | | | | | | |
| -0.4 | | | | | | | | | | | | | | |
| -0.35 | | | | | | | | | | | | | | |
| -0.3 | | | | | | | | | | | | | | |
| -0.2 | | | | | | | | | | | | | | |
| -0.1 | | | | | | | | | | | | | | |
| 0 | | | | | | | | | | | | | | |
| 0.1 | | | | | | | | | | | | | | |
| 0.2 | | | | | | | | | | | | | | |
| 0.3 | | | | | | | | | | | | | | |
| 0.35 | | | | | | | | | | | | | | |
| 0.4 | | | | | | | | | | | | | | |
| 0.5 | | | | | | | | | | | | | | |
| 0.6 | | | | | | | | | | | | | | |
| 0.7 | | | | | | | | | | | | | | |
| 0.8 | | | | | | | | | | | | | | |
| 0.832 | | | | | | | | | | | | | | |
| 1 | | | | | | | | | | | | | | |
| 1.2 | | | | | | | | | | | | | | |
| 1.4 | | | | | | | | | | | | | | |
| 1.5 | | | | | | | | | | | | | | |
| 1.6 | | | | | | | | | | | | | | |
| 2 | | | | | | | | | | | | | | |
| 2.5 | | | | | | | | | | | | | | |
| 3 | | | | | | | | | | | | | | |

**Figure A3.** Measurement plan for the axial evaluation of all rotor position sensors; blue, resolvers; green, inductive sensors; orange, AMR sensor.

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
