# Peer review of "Benchmark of Rotor Position Sensor Technologies for Application in Automotive Electric Drive Trains"

_electronics, doi:10.3390/electronics9071063_

Round 1
Reviewer 1 Report
The paper is well written and of general interest. I have only minor remarks:
- In Eq. (4) there is atan2 function, in Eq. (9) there is arctan (in italics) and in (13) atan, not in italics. It would be preferable to use only one function designation, I recommend arctan (or arctan2), not in italics (it is a function, not a variable).
- Equations should be written with full size fractions, not like atan(Vsin(t)/Vcos(t)).
- Differentials in equations should be written not in italics (eq. (5), (14),
- There should be spaces between numerical values and units (lines 344, 345, 428, 444, for example).
Some spell check would be preferable.
Reviewer 2 Report
- How can the angular information of the rotating magnet be determined by measuring the bridges’ resistance? Please explain clearly and provide the theorem. That is the Eq.(10) must be proved.
- The comparision of methods of AMR position sensors and GMR sensors should be presented.
- Fig.16-18 are demonstrated as bar chart, how can authors prove the results are verified? I suggest you to add some real time process diagrams to justify the good performances.
